# *Aphelinus nigritus* Howard (Hymenoptera: Aphelinidae) Preference for Sorghum Aphid, *Melanaphis sorghi* (Theobald, 1904) (Hemiptera: Aphididae), Honeydew Is Stronger in Johnson Grass, *Sorghum halepense*, Than in Grain Sorghum, *Sorghum bicolor*

**DOI:** 10.3390/insects14010010

**Published:** 2022-12-22

**Authors:** Crys Wright, Anjel M. Helms, Julio S. Bernal, John M. Grunseich, Raul F. Medina

**Affiliations:** Department of Entomology, Texas A&M University, TAMU 2475, College Station, TX 77843-2475, USA

**Keywords:** contact kairomones, honeydew, parasitoids, chemical ecology, plant-insect interactions

## Abstract

**Simple Summary:**

Like many aphids, the sorghum aphid (SA) *Melanaphis sorghi* produces honeydew, a waste product that can increase parasitoid retention and elicit foraging behaviors on plants. We determined the potential of SA honeydew to retain a common aphid parasitoid, *Aphelinus nigritus*, to assess the practicality of SA biological control on grain sorghum, *Sorghum bicolor*. Since SAs also feed on Johnson grass, *Sorghum halepense*, an alternative plant host, we characterized the composition of honeydew from aphids feeding on either grain sorghum or Johnson grass and evaluated *A. nigritus* preference for honeydew produced on either plant species. Our study found that *A. nigritus* often remained on honeydew produced by SAs feeding on both grain sorghum and Johnson grass. However, despite sharing similar sugar, amino acid, and organic acid profiles, honeydew produced on Johnson grass was preferred by *A. nigritus* over honeydew produced on grain sorghum. Our results suggest that SA honeydew could facilitate *A. nigritus* parasitoid retention on Johnson grass to lower SA populations before the grain sorghum growing season.

**Abstract:**

How aphid parasitoids of recent invasive species interact with their hosts can affect the feasibility of biological control. In this study, we focus on a recent invasive pest of US grain sorghum, *Sorghum bicolor*, the sorghum aphid (SA), *Melanaphis sorghi*. Understanding this pest’s ecology in the grain sorghum agroecosystem is critical to develop effective control strategies. As parasitoids often use aphid honeydew as a sugar resource, and honeydew is known to mediate parasitoid–aphid interactions, we investigated the ability of SA honeydew to retain the parasitoid *Aphelinus nigritus*. Since SAs in the US have multiple plant hosts, and host–plant diet can modulate parasitoid retention (a major component in host foraging), we measured SA honeydew sugar, organic acid, and amino acid profiles, then assessed via retention time *A. nigritus* preference for honeydew produced on grain sorghum or Johnson grass, *Sorghum halepense*. Compared to a water control, *A. nigritus* spent more time on SA honeydew produced on either host plant. Despite similar honeydew profiles from both plant species, *A. nigritus* preferred honeydew produced on Johnson grass. Our results suggest the potential for SA honeydew to facilitate augmentation strategies aimed at maintaining *A. nigritus* on Johnson grass to suppress SAs before grain sorghum is planted.

## 1. Introduction

Parasitoids locate their hosts using a variety of visual, contact, and olfactory cues. In the context of tri-trophic interactions, parasitoid success in finding their hosts is largely influenced by the herbivore’s host plant [1,2,3]. That is because plants contain physical [4] and chemical characteristics [5] that can attract or repel foraging parasitoids [6]. Host plants can alter parasitoid searching strategy, especially in cases where the same insect host feeds on multiple host plants. Polyphagy in insect hosts can lead to differential parasitoid attack when insect hosts feed on different host plants [6,7,8,9]. Among insect hosts, aphids, whose limited mobility make them highly susceptible to parasitism [10,11,12], produce honeydew, a sugar-rich waste product that can nutritionally supplement parasitoids [13,14]. Since many aphids feed on multiple host plants, plant-mediated differences in honeydew may affect parasitoid retention on this sugar source, as is seen in *Encarsia formosa* parasitoids [15]. Overall, the potential for honeydew-mediated retention of parasitoids, a large component in host-searching and parasitism behaviors, is understudied and warrants attention.

Many species of parasitoids require sugar-rich diets to meet metabolic needs. In environments rich with floral resources, nectar is a well-studied and highly nutritious source of food [16,17,18]. Comparatively, fewer studies have assessed the suitability of honeydew, a sugar-rich waste product of aphids and other hemipterans, as a food alternative. Honeydew, which comprises carbohydrates, amino acids, and organic acids [19,20,21,22,23], serves as a food source for some parasitoids [24,25,26,27,28], and predators [27], along with other organisms like ant mutualists [29], inhabiting low-nectar environments. Honeydew utilization is especially common in parasitoids of cereal aphids, even in the presence of nectar sources [30], likely because parasitoids reduce energy-use costs by remaining within a host patch when consuming honeydew instead of actively searching for flowers [27,30].

In addition to a nutritional source, honeydew is considered a kairomone, or a semiochemical host by-product that benefits another species. Aphid by-products, such as honeydew, along with glandular [31] and cornicle secretions [32], are key determinants of parasitoid retention behaviors [33] that increase the efficiency of host location and can improve the likelihood of parasitism [13,14,34]. As a contact kairomone, honeydew can elicit parasitoid behaviors in the absence of aphids and other hemipterans. For example, *Psyllaephagus pistaciae* parasitoids increase their searching behavior in the presence of psyllid honeydew alone [35]. As honeydew can retain parasitoids within infested plants and provide an important food resource, the role of honeydew in parasitoid-mediated aphid suppression merits investigation.

The sorghum aphid (SA), *Melanaphis sorghi*, originally reported as the sugarcane aphid, *Melanaphis sacchari* (Zehntner, 1897), is a cereal pest of economic concern in Texas and has become a serious pest of grain sorghum, *Sorghum bicolor* [36]. Although some studies have documented a combination of management methods and production practices to manage this pest in grain sorghum [37,38,39], very little is known about the role of natural enemies of SA [37]. SA is currently attacked by a variety of predators and parasitoids in the southern US, including the generalist parasitoid *Aphelinus nigritus*. *A. nigritus* females are synovigenic and feed on host hemolymph to complete egg maturation (Hopper, University of Delaware, pers. communication). When feeding, SAs produce copious amounts of honeydew on grain sorghum, which may provide a potential food resource to this parasitoid. Since parasitoids can also use honeydew as a contact kairomone, SA honeydew might lead to *A. nigritus* retention and facilitate parasitism. Overall, understanding the degree of *A. nigritus* retention on SA honeydew will increase knowledge of the dynamics facilitating parasitism in this parasitoid–host system.

Whereas most honeydews contain a range of sugars, amino acids, and organic acids [26,40,41,42], the composition and concentration of these macronutrients vary among hemipterans, host plant species, and over time [24,43]. In the case of SA, honeydew varies among cultivars of sugarcane, *Saccharum* spp. [44]. Honeydew variation is important because it can influence parasitoid preference, as seen with *Aphidius ervi*, which prefers honeydew from the bird cherry-oat aphid, *Rhopalosiphum padi*, over that of the English grain aphid, *Sitobion avenae*, and green peach aphid, *Myzus persicae*, even when all species feed on the same host plant [26]. In the case of oligophagous or polyphagous insect hosts, parasitoids may prefer honeydew produced by the same aphid species feeding on different plants [45,46,47]. Honeydew quantity can also influence parasitoid foraging [25], likely because it reflects the population densities of aphid hosts [30]. In agroecosystems where insects deposit honeydew on both crops and weedy vegetation, differences in honeydew quality and quantity could lead parasitoids to attack insect hosts more in one plant over another, resulting in skewed parasitism rates.

SA also feeds on the perennial weed Johnson grass, *Sorghum halepense* [48], an uncultivated weed abundant across the southern US [49] that frequently occurs in proximity to grain sorghum. Notably, rates of SA parasitism are reportedly greater on grain sorghum versus Johnson grass (B. Elkins, unpublished data). As host-plant identity largely determines honeydew composition [43], and differences in honeydew may affect parasitoid preference, plant-derived differences in SA honeydew quality and/or quantity could facilitate differences in parasitoid retention and lead to differences in parasitism rates. A better understanding of variations in SA honeydew across host plant species and over time may help determine how parasitoids and other natural enemies modulate their selection of hosts.

In this study, sugar, amino acid, and organic acid compositions and concentrations of honeydew from aphids feeding on grain sorghum or Johnson grass were compared. Subsequently, *A. nigritus* retention on honeydew from aphids feeding on these two host plants was assessed at different collection timepoints (i.e., after honeydew accumulated for 24, 72, or 120 h). As shown in other studies, we expected to detect differences in SA honeydew composition between host plants and over time. Considering the trend of less parasitism in Johnson grass, we also predicted a strong *A. nigritus* preference for SA honeydew produced on grain sorghum. Finally, and since parasitoid foraging generally increases with larger quantities of honeydew [30,50], we further predicted a stronger *A. nigritus* response to SA honeydew produced on grain sorghum when collected over a period of 120 versus 24 h.

## 2. Materials and Methods

### 2.1. Establishing Main Aphid Colonies

SAs were reared on either DEKALB^®^ DKS 4420 (Bayer, St. Louis, MO, USA), a susceptible grain sorghum variety or wild Johnson grass collected from the Texas A&M Research Farm in Sommerville, TX (30°31′54.8″ N 96°25′50.2″ W). Hence, 3–5 sorghum seeds were planted in 3.8 cm diameter × 21 cm high planting tubes (Amazon, Seattle, WA, USA) containing Sun Gro^®^ Metro-Mix^®^ 360 (Sun Gro^®^ Horticulture, Agawam, MA, USA). The rhizomes of field collected Johnson grass were cut, washed in 2:100 volume of soap and water and replanted in planting tubes. Plants were grown at 27 °C under a 16L:8D cycle and 70% relative humidity for three weeks.

Aphids were collected from grain sorghum (30°32′33.0″ N 96°25′36.4″ W) and Johnson grass (30°32′18.3″ N 96°25′04.4″ W) field sites and placed on respective grain sorghum or Johnson grass plants in separate 40 × 30 × 30 cm cages to establish main colonies. Main colonies were defined as those containing a genetic mix of SAs. Cages were constructed from plexiglass sheets (ACME Glass Company, Bryan, TX, USA) and fused together with methylene chloride and masking tape. Grain sorghum and Johnson grass-reared SA colonies were maintained under the conditions listed above. Plants were added to both colonies each week to sustain population numbers.

### 2.2. Establishing Clonal Aphid Colonies

Aphid clonal colony cages were constructed from one-liter plastic bottles. A three-mm diameter hole was cut around the neck of the bottle, while the bottom portion was completely removed. The base of a plant tube was then fitted through the hole and secured with masking tape. To prevent aphid escape, the bottle was covered with nylon hosiery. With a paintbrush, a single apterous aphid from the main SA colonies generated above was placed on a grain sorghum or Johnson grass leaf in each bottle cage. Aphids were placed on the same plant species from which they were reared in the main colonies. In total, 20 clonal colonies per plant species were maintained in a rearing room under the same conditions mentioned above.

### 2.3. Rearing Parasitoids

Grain sorghum or Johnson grass plants infested with SA from the main colonies were placed in separate 40 × 30 × 30 cm cages to rear *A. nigritus*. These *A. nigritus* cages were put in a different room (to avoid parasitoid contamination of the main colonies), under similar light and temperature conditions as the SA main colonies above. *A. nigritus* mummies (successfully parasitized SA) were obtained from the grain sorghum and Johnson grass field sites referenced above and individually placed in 0.2 mL PCR tubes (Thermo Fisher Scientific, Waltham, MA, USA). Aphid mummies were observed daily until parasitoid emergence. All emerged *A. nigritus* were transferred to *A. nigritus* cages containing grain sorghum or Johnson grass SA-infested plants. Mummies produced in either parasitoid cage were placed in separate PCR tubes and monitored daily until emergence. Only female parasitoids aged 0–24 h were used in experiments. Once emerged, females were transferred to separate 1.5 mL centrifuge tubes (VWR International, Radnor, PA, USA) containing two μL of autoclaved water (smeared on the sides of the tube to allow hydration while preventing drowning) and a 0–72 h-old male. Females and males were observed until mating occurred.

### 2.4. Honeydew Collection

Ten apterous, adult aphids from each of the 20 grain sorghum and Johnson grass clonal colonies were transferred via paintbrush to separate clip cages. Clip cages were lined with round plastic disks for honeydew droplet collection and attached to the leaves of three-week-old grain sorghum or Johnson grass plants. Aphids were placed on the same plant species from which they were reared. The cages were clipped to each plant and honeydew was deposited continuously for one of three timepoints: 24, 72, or 120 h. After freeze-drying for 24 h, the honeydew disks were weighed, then diluted in 15 μL High Pressure Liquid Chromatography (HPLC)-grade water (Sigma-Aldrich, St. Louis, MO, USA) filtered through spin columns (Thermo Fisher Scientific, Waltham, MA, USA) and stored in 1.5 mL centrifuge tubes at −20 °C until use. Twenty replicates (clip cages) per plant species and timepoint were used for HPLC analysis, while 20 additional replicates were used in bioassays to assess parasitoid preference.

### 2.5. Analysis of Honeydew Composition Using HPLC

Sugar and organic acid composition of SA honeydew was analyzed using an Agilent 1200 HPLC connected to a 1260 Infinity II refractive index detector (Agilent Technologies, Santa Clara, CA, USA). Samples were thawed and sonicated for two minutes, then spun down in a centrifuge at 13,500 rpm. Samples were then transferred to two ml screw top vials containing 150 μL glass inserts (Agilent Technologies, Santa Clara, CA, USA) and run on the Agilent 1200 binary LC gradient system using the Hi-Plex Ca (Duo) 7.7 × 50 mm column and 8 μL guard column (Agilent Technologies, Santa Clara, CA, USA). The column was eluted with 100% HPLC water at a flow rate of 4.0 mL/min at 80 °C [51]. The amino acid composition of SA honeydew was analyzed using an Agilent 1200 HPLC connected to a diode array detector (Agilent Technologies, Santa Clara, CA, USA). Samples were separated using the AdvanceBio AAA 2.7 μm 4.6 × 100 mm column with a 2.7 μm guard column (Agilent Technologies, Santa Clara, CA, USA). Two eluent mixtures (A:10 mM Na2HPO4–10 mM Na2B4O7 at a pH of 8.2 and B: 45:45:10% ACN: MeOH: HPLC Grade water) were run through the column at a flow rate of 1.2 mL/min at 40 °C. Prior to injection, amino acids were derivatized with o-phthalaldehyde (OPA), 9-fluorenylmethyloxycarbonyl (FMOC), a borate buffer, and injection diluent containing 100 mL of eluent mixture A and 0.4 mL concentrated H3PO4 (Agilent Technologies, Santa Clara, CA, USA) to create fluorescent molecules for enhanced detection at the 390 nm DAD wavelength [52]. Using the Agilent OpenLab ChemStation software, sugar, organic acid, and amino acid identities were determined by comparing retention times to those of authentic standards (Sigma-Aldrich, St. Louis, MO, USA). Quantities for each compound were determined by comparing peak areas to calibration curves.

### 2.6. Statistical Analyses of Honeydew Composition

Statistical analyses were performed using R, Version 4.0.5 (R Core Team, Vienna, Austria). Using the R-package VEGAN [53], honeydew sugar and amino acid content were analyzed by conducting permutational multivariate analysis of variance (PERMANOVA) to quantify differences in blends [54] and non-metric multidimensional scaling ordinations to visualize blend differences at different collection timepoints and between host plants [54]. The normality of the data was verified using Levene’s test of equality of variances. An ANOVA assessed the effects of collection timepoint, host plant diet, and an interaction between the two on the total sugar concentration of honeydew. Total sugar concentration of honeydew was the dependent variable, while collection timepoint and host plant diet were independent variables. Tukey’s HSD assessed pairwise comparisons in the total sugar concentration of honeydew by host plant and between each collection timepoint. One-way ANOVAs were used to compare individual sugar, amino acid, and organic acid compounds between host plants within collection timepoints.

### 2.7. Measuring Parasitoid Preference

Parasitoid preference for either honeydew source was measured at two timepoints: 24 and 120 h. Hence, 5 μL of honeydew collected after 24 and 120 h (from each of the 20-grain sorghum and Johnson grass replicate plants mentioned above) were pipetted onto respective two-week old grain sorghum and Johnson grass leaves. The leaves were then placed on opposite sides of a 30 mm diameter × 11 mm high Petri dish. With a paintbrush, one female parasitoid was placed at the center of the dish. The parasitoid was allowed to acclimate for five minutes before recording behavior for 10 min using the Behavioral Observation Research Interactive Software (BORIS, Turin, PIE, Italy) [55]. Preference was measured as the relative proportion of time spent (relative retention) on each treatment (=total time on one treatment leaf/the total time spent on both treatment leaves). To test parasitoid retention on either honeydew source, grain sorghum or Johnson grass leaves with a drop of water were used as no-sugar sources in a choice-test setting. The study contained three choice test variations: (1) honeydew produced by SAs feeding on grain sorghum (herein referred to as grain sorghum honeydew) versus water on a grain sorghum leaf, (2) honeydew produced by SAs feeding on Johnson grass (herein referred to as Johnson grass honeydew) versus water on a Johnson grass leaf, and (3) grain sorghum honeydew versus Johnson grass honeydew on their respective leaves. Accounting for bias toward the original parasitoid aphid host (i.e., whether mothers of experimental parasitoids were reared on grain sorghum or Johnson grass fed SAs), each choice test variation per timepoint consisted of 20 replicates, 10 with parasitoids whose mothers were reared on grain sorghum-fed SAs, and the other 10 with parasitoids whose mothers were reared on Johnson grass-fed SAs. To avoid directional bias, the locations of plant leaves on the Petri dish were swapped every other replicate. Each replicate consisted of a separate female parasitoid.

### 2.8. Statistical Analyses of Parasitoid Preference

Statistical analyses were performed using JMP^®^, Version 15.2 (SAS Institute Inc., Cary, NC, USA). Data from both honeydew versus water experiments (grain sorghum honeydew versus water and Johnson grass honeydew versus water) at both collection timepoints (24 and 120 h) were compared to assess the strength of retention on either honeydew when a second honeydew source was not present. Preference for honeydew as well as by the original parasitoid aphid host and collection timepoint was assessed in an ANCOVA. The effect of two available sources of honeydew (grain sorghum versus Johnson grass) on parasitoid preference, along with collection timepoint and original parasitoid aphid host was measured in a separate ANCOVA. For the first ANCOVA comparison, the relative proportion of time was measured as the total time spent on grain sorghum honeydew (or Johnson grass honeydew)/the total time spent on each honeydew source and water. For the second ANCOVA, the relative proportion of time was measured as the total time spent on grain sorghum honeydew/the total time spent on both grain sorghum and Johnson grass honeydews. The data were converted to arcsine square-root values (ASIN (SQRT ×)) × 57.296) and considered as the dependent variable, while collection timepoint, original parasitoid aphid host (i.e., parasitoids whose mothers were reared on grain sorghum or Johnson grass fed SAs) were independent variables. Choice treatment (grain sorghum honeydew, Johnson grass honeydew, or water) was also factored to assess the strength of retention on one honeydew source over another, between honeydew versus water experiments. Honeydew concentration (dry weight in μg/15 μL HPLC-grade water) was used as a covariate to account for differences in concentration between 24- and 120-h and grain sorghum and Johnson grass honeydew samples. One-way ANOVAs were then conducted to compare parasitoid preference within treatments, by collection timepoint, and by original parasitoid aphid host (null hypothesis = no difference in relative time spent on one choice treatment versus another).

## 3. Results

### 3.1. SAs Fed on Grain Sorghum and Johnson Grass Excrete Varied Concentrations of Honeydew

The total sugar concentration of honeydew from SAs feeding on grain sorghum and Johnson grass significantly differed by timepoint and by host plant, but there was no significant interaction between the two (Table 1). Overall, sugar concentration was greater in Johnson grass samples (F_2,104_ = 6.0142, *p* = 0.0159) (Figure 1A). Total sugar concentrations increased over time, although sugar concentration did not vary between the 72 and 120-h timepoints (Figure 1B). Despite this, there was a trend towards greater concentrations of sugar in Johnson grass versus grain sorghum at each timepoint.

### 3.2. Sugar, Amino Acid, and Organic Acid Profiles Are Similar between Host Plants

Sugar and organic acid profiles were not significantly different between grain sorghum and Johnson grass honeydew collected after 24 (PERMANOVA F_1,34_ = 3.0852, *p* = 0.06) (Figure 2A) and 72 (PERMANOVA F_1,34_ = 1.5388, *p* = 0.212) hours (Figure 2B). There was a significant difference in profiles between honeydews collected after 120 h (PERMANOVA F_1,34_ = 3.7925, *p* = 0.01) (Figure 2C). The sugars detected in grain sorghum and Johnson grass honeydew samples are shown in Table 2. Fructose was marginally more abundant in Johnson grass honeydew than in grain sorghum honeydew after 72 h (F_1,34_ = 4.215, *p* = 0.0474). The only organic acid detected in SA honeydew was fumaric acid, which was more abundant in Johnson grass than in grain sorghum honeydew at both 24 (F_1,34_ = 10.71, *p* = 0.0025) and 120-h (F_1,34_ = 5.325, *p* = 0.0272) collection timepoints (Table 2). Amino acid profiles were not significantly different between host plant honeydews at any of the collection timepoints. Of the detected amino acids, proline, serine, aspartic, and glutamic acid were most abundant on honeydew samples in both plant species. Host plant-specific differences in amino acids were observed for tryptophan after 120 h (F_1,34_ = 6.04, *p* = 0.0198) and Tyrosine after 24 h (F_1,34_ = 9.401, *p* = 0.0041) (Appendix A).

### 3.3. Aphelinus Nigritus Retention Behavior Observed on Honeydew Excreted by Aphids Feeding on Both Grain Sorghum and Johnson Grass

Across treatments, parasitoid preference (measured as the relative proportion of time spent on each treatment) was not influenced by collection timepoint or by the aphid host from which the parasitoid emerged. In comparing the two experiments, parasitoids preferred honeydew over water, no matter on which host plant honeydew was provided (Table 3). When given the choice between grain sorghum honeydew and water, both grain sorghum and Johnson grass parasitoids preferred grain sorghum honeydew at both timepoints (Figure 3A). Given the choice between Johnson grass honeydew and water, grain sorghum parasitoids preferred Johnson grass honeydew at both 24-h and 120-h timepoints. However, Johnson grass parasitoids preferred Johnson grass honeydew at the 24-h timepoint but made no choice at the 120-h timepoint (Figure 3B).

### 3.4. Aphelinus Nigritus Prefers Johnson Grass over Grain Sorghum Honeydew

Across treatments, preference was not influenced by collection timepoint, by the aphid host from which the parasitoid emerged, or by an interaction between the two (Table 4). At the 24-h timepoint, grain sorghum parasitoids preferred Johnson grass honeydew, while Johnson grass parasitoids preferred grain sorghum honeydew. At 120 h, both grain sorghum and Johnson grass parasitoids preferred Johnson grass honeydew (Figure 4).

## 4. Discussion

*Aphelinus nigritus* preferred SA honeydew produced by SAs feeding on Johnson grass over that produced on grain sorghum. This preference does not appear to be mediated by differences in honeydew sugar, amino acid, or organic acid composition between honeydews from SAs feeding on Johnson grass or grain sorghum. The sugar and organic acid profiles were mostly similar between honeydews produced by SAs feeding on Johnson grass and grain sorghum, except for honeydews collected after 120 h. Differences at this timepoint appear to stem from a greater accumulation of fumaric acid in Johnson grass honeydew. No study to our knowledge has assessed the role of fumaric acid in parasitoid retention. Consequently, it is unknown as to whether this organic acid affects parasitoid foraging on Johnson grass honeydew. Amino acid profiles were similar between host plants across all timepoints. Approximately 70% of all detected amino acids consisted of aspartic acid, glutamic acid, proline, and serine, all nonessential amino acids for aphids [21,26,56], which obtain essential amino acids from the obligate bacterial endosymbiont *Buchnera aphidicola* [57]. Further, aspartic acid, glutamic acid, and proline are likely not limiting amino acids, having all been detected in high amounts on several cultivars of grain sorghum [58,59] and detected in Johnson grass honeydew samples in this study.

We observed a higher concentration of total honeydew sugars in Johnson grass. Higher honeydew sugar concentrations may result from generally higher concentrations of sugar in Johnson grass tissues. However, despite studies separately assessing sugar concentration in Johnson grass rhizomes and leaves [60,61] and grain sorghum kernels [62], direct measurements of differences in phloem sugar concentration between these two plant species have not been attempted, to our knowledge. Future comparisons of phloem sugar concentration between plant species might help explain our results. If sugar concentration is greater in grain sorghum plants, a higher sugar concentration in aphid honeydew feeding on Johnson grass could result from compensatory aphid feeding. Aphids can adjust their dietary intake based on nutritional (mainly nitrogen) requirements [63] and nutrient supply [64,65]. It is possible that SAs consumed more Johnson grass phloem, leading to higher honeydew sugar concentration, to meet their nitrogen needs. In addition to measuring phloem sugars, this hypothesis could be validated by measuring phloem amino acid content (a major source of nitrogen) and using electrical penetration graphs to measure SA feeding duration and frequency on both plant species.

Through the honeydew preference bioassays, it was evident that honeydew from either plant species led to *A. nigritus* retention. In the honeydew versus water choice tests, honeydew from aphids fed on the two host plants tested elicited *A. nigritus* arrestment, feeding, and searching behaviors. The specific sugar requirements of this parasitoid have not been reported. However, several aphelinids readily consume and display searching behaviors on host honeydew [15,66,67], suggesting that *A. nigritus* may use honeydew as an SA foraging cue. This is not always common in parasitoids, likely due to high levels of crystallized oligosaccharide sugars that may reduce honeydew palatability [68]. Along with crystallization, high viscosity can also make honeydew sugars difficult to consume [24], which is why parasitoids tend to prefer less viscous nectar resources [69]. However, during SA infestations, SA honeydew is likely the most reliable and widely available sugar resource for *A. nigritus* inhabiting grain sorghum and Johnson grass fields. Even if nectar or extra floral nectar are available along field margins, *A. nigritus* may choose the sugar option closest to their host and present in high abundances, as seen in other parasitoids [30]. The decision to consume sugar within host patches likely reduces energy use and host searching costs, creating selective pressure to evolve honeydew consumption instead of relying on nectar and the associated active searching for flowers [70,71].

The original parasitoid aphid host did not generally affect this parasitoid’s preference for an available sugar resource. While sugar requirements specific to *A. nigritus* have not been reported, species of Aphelinidae are known to consume glucose, fructose, and sucrose [67,72,73], which were the three most abundant sugars in SA honeydew from either host plant species. It is important to remember that like most parasitoids, *A. nigritus* would require some source of sugar to maintain enough energy for foraging [74]. Thus, females would likely not reject available honeydew, irrespective of the host plant, as corroborated by our honeydew versus water choice tests, where parasitoids displayed similarly strong preferences for either grain sorghum or Johnson grass honeydew over water.

When given a choice between grain sorghum and Johnson grass honeydew, most parasitoids preferred Johnson grass honeydew. This is surprising as it directly contrasts with field observations of higher *A. nigritus* parasitism rates on grain sorghum (B. Elkins, unpublished data) and suggest that *A. nigritus* honeydew preferences may not align with their host oviposition preference. Since the factors affecting field parasitism rates are complex in nature, differing *A. nigritus* parasitism rates may reflect physical instead of chemical plant traits. Early-stage grain sorghum and Johnson grass plants (like the three-week-old plants used in this study) are similar in texture, and in preliminary trials measuring *A. nigritus* preference for water present on grain sorghum and Johnson grass leaves, we found no inherent bias towards leaf type (data not shown). However, compared to grain sorghum, mature Johnson grass has more bicellular trichomes that can lead to microroughness on the leaves [75]. Leaf surface can influence parasitism rates of the same host feeding on multiple plants. For example, Mulatu et al. [76] observed lower instances of parasitism in the potato tuber moth, *Phthorimaea operculella*, when feeding on tomato versus potato leaves. This is attributed to a high density of glandular trichomes on tomato that deter parasitoid visitation. Differences in leaf surface may also significantly slow parasitoid movement, which could make finding hosts difficult [77]. Moreover, our study used naïve parasitoids. As a result, we did not consider parasitoid learning, a behavior that can influence host preference [78] and may contribute to observed *A. nigritus* parasitism rates.

The only exception to *A. nigritus*’s preference for Johnson grass honeydew were parasitoids originating from Johnson grass, which preferred grain sorghum over Johnson grass honeydew collected after 24 h. This result, along with the observations of no preference in Johnson grass parasitoids between Johnson grass honeydew collected after 120 h and water are perplexing. Since honeydew is a suitable medium for microbial growth [23,79,80], these results were initially presumed to stem from differences in the microbial compositions of grain sorghum and Johnson grass honeydews. Available data on the microbial compositions of both honeydews at different timepoints (J. Holt, unpublished data) does not reveal evidence of any correlation between microbial composition and parasitoid preferences for honeydew as measured in this study. We hypothesize that the variation in parasitoid honeydew preference may not correlate with the presence of microbial organisms themselves, but rather with the differential presence of microbial metabolites or microbial volatile organic compounds (MVOCs) in honeydew from different host-plant species, which are both shown to mediate natural enemy activity. For example, some metabolites of the bacterium *Bacillus thuringiensis* are known to repel and deter oviposition of *Eretmocerus eremicus* parasitoids on whiteflies [81]. With respect to volatiles, Fand et al. [82] isolated VOC-producing bacteria from grapevine mealybug honeydew that was highly attractive to the endoparasitoid *Anagyrus dactylopii*. Similarly, Leroy et al. [23] isolated microorganisms’ volatiles from pea aphid honeydew that attracted and enhanced predation by hoverflies. Based on our results, the microbial metabolites or MVOCs present in SA honeydew fed on different diets should be assessed and their role in mediating *A. nigritus* honeydew preferences tested.

Despite an apparent host preference for SAs feeding on grain sorghum, *A. nigritus* retention on Johnson grass SA honeydew has positive implications for future honeydew-related pest management, as it suggests that Johnson grass honeydew can maintain parasitoids on SA-infested plants. In the long term, this might lead to the augmentation of parasitoid populations to potentially suppress SA on Johnson grass before it spills over to grain sorghum during the crop’s growing season. Whether *A. nigritus* honeydew retention can lead to SA suppression is another relevant question that should inform any future assessments of this parasitoid’s potential as a biocontrol agent of SA.

## Figures and Tables

**Figure 1 insects-14-00010-f001:**
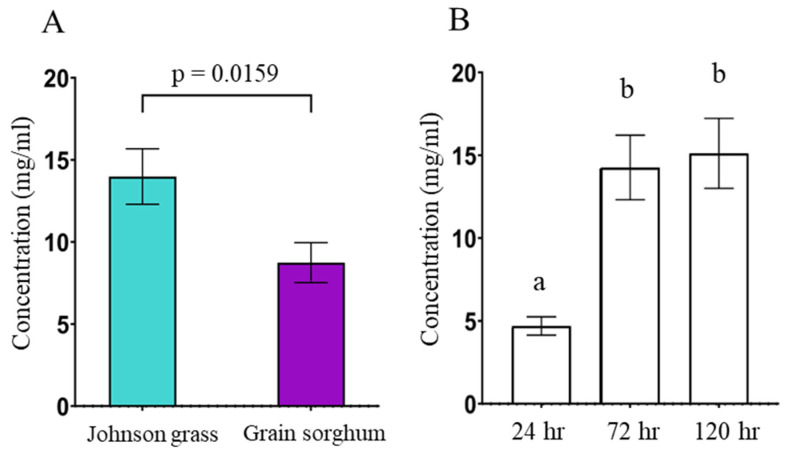
(**A**) Overall, sugar concentration is greater in honeydew from Johnson grass aphids (see Table 1). Statistical significance at *p* < 0.05. (**B**) Total sugar concentration of SA honeydew was significantly lower at the 24-h timepoint but no significant difference in sugar concentration was detected between the 72 and 120-hr timepoints. As per Tukey’s HSD. Different letters indicate statistical significance across timepoints at *p* < 0.05.

**Figure 2 insects-14-00010-f002:**
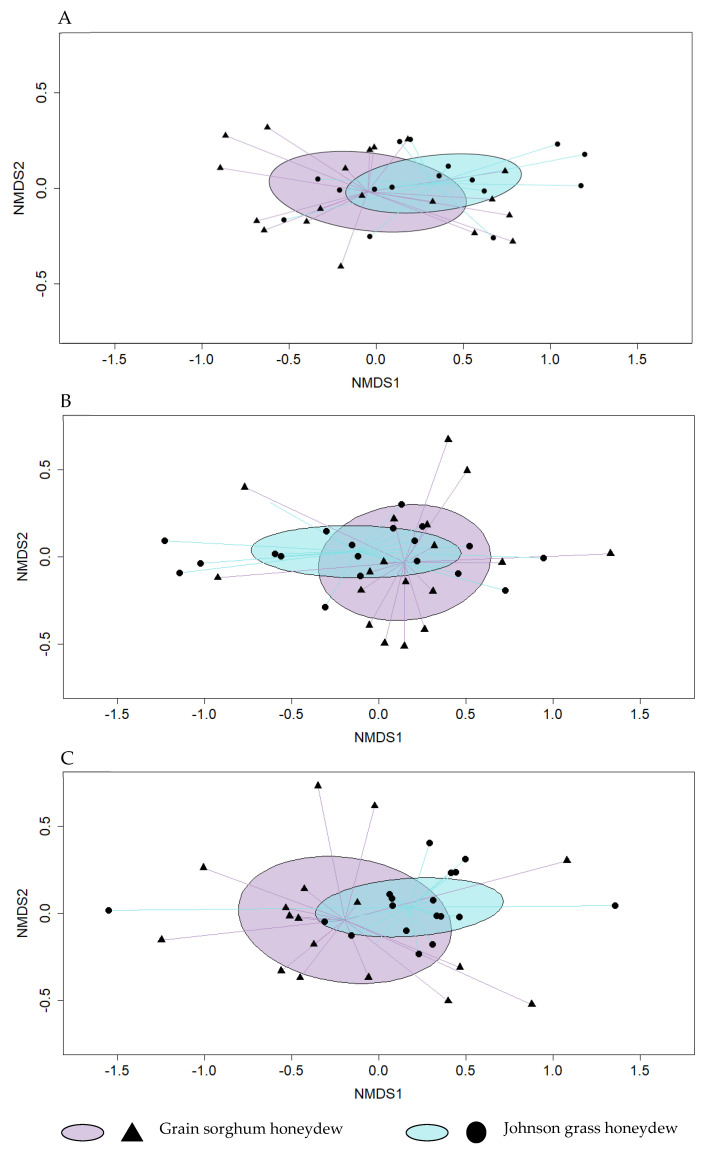
Non-metric multidimensional scaling (NMDS) ordination diagram. Sugar and organic acid profiles are similar between grain sorghum and Johnson grass honeydew collected after 24 h (**A**) and 72 h (**B**). Profiles are significantly different between grain sorghum and Johnson grass honeydew collected after 120 h (PERMANOVA F_1,34_ = 3.7925, *p* = 0.01) (C). Ellipses represent 95% confidence intervals. Triangle and circle points ordinated closer to one another are more similar.

**Figure 3 insects-14-00010-f003:**
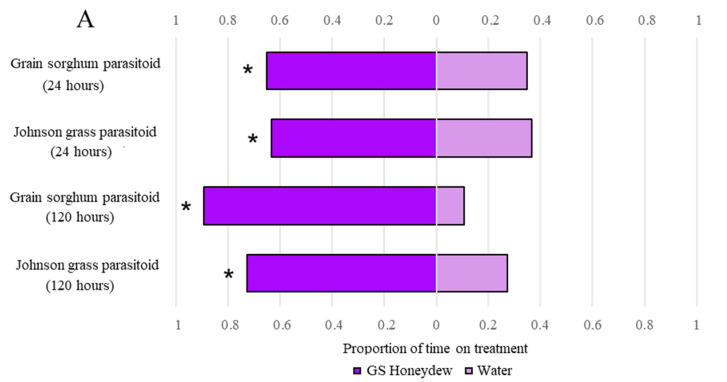
(**A**) Grain sorghum and Johnson grass parasitoid retention is greater on grain sorghum honeydew. (**B**) Grain sorghum and Johnson grass parasitoid retention is largely greater on Johnson grass honeydew. As per a one-sample *t*-test, an asterisk indicates statistical significance at *p* < 0.05.

**Figure 4 insects-14-00010-f004:**
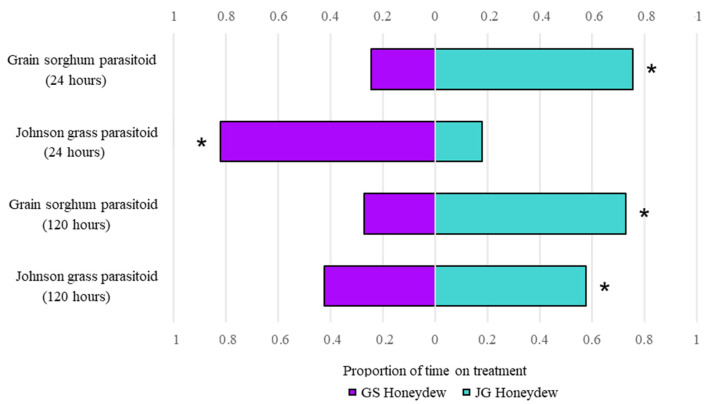
Both grain sorghum and Johnson grass parasitoids preferred Johnson grass honeydew, except for Johnson grass parasitoids at the 24-h timepoint (one-sample *t*-test). As per a one-sample *t*-test, an asterisk indicates statistical significance at *p* < 0.05.

**Table 1 insects-14-00010-t001:** Results from ANOVA Indicate that Timepoint and Host Plant Diet Affect Honeydew Total Sugar Concentration. Bolded Values Indicate Statistical Significance at *p* < 0.05.

Variables	F Value	df	*p* Value
Collection Timepoint	11.2749	2.104	**<0.0001**
Host Plant	6.0142	2.104	**0.0159**
Collection Timepoint × Host Plant	0.4947	2.104	0.6112

**Table 2 insects-14-00010-t002:** Sugar and Organic Acid Concentrations (mg/mL) of Grain sorghum and Johnson grass SA Honeydew over Time. Values Are Mean ± SE of the Mean. As per One-way ANOVA, Bolded Values within Timepoints Indicate Statistical Significance at *p* < 0.05.

	24 h		72 h		120 h	
Sugar/ Organic Acid	Grain Sorghum	Johnson Grass	Grain Sorghum	Johnson Grass	Grain Sorghum	Johnson Grass
Fructose	0.66 ± 0.71	1.18 ± 0.73	2.47 ± 0.71	**4.67 ± 0.68**	3.32 ± 0.73	4.87 ± 0.70
Glucose	0.76 ± 0.66	1.3 ± 0.68	2.39 ± 0.66	4.32 ± 0.63	2.91 ± 0.68	4.55 ± 0.64
Melezitose	0.1 ± 0.10	0.11 ± 0.10	0.39 ± 0.10	0.43 ± 0.10	0.35 ± 0.12	0.52 ± 0.10
Stachyose	0.02 ± 0.01	0.02 ± 0.01	0.03 ± 0.01	0.02 ± 0.01	0.01 ± 0.01	0.01 ± 0.01
Sucrose	1.36 ± 0.99	1.34 ± 0.96	3.47 ± 0.96	4.58 ± 0.89	3.15 ± 0.96	4.17 ± 0.91
Fumaric Acid	1.16 ± 0.35	**1.82 ± 0.37**	2.39 ± 0.36	3.24 ± 0.34	2.34 ± 0.37	**3.76 ± 0.35**

**Table 3 insects-14-00010-t003:** Neither Timepoint nor Original Parasitoid Aphid Host Mediated Parasitoid Preference (Measured as the Relative Proportion of Time Spent on Each Treatment) for Grain Sorghum or Johnson Grass Honeydew versus Water. Bolded Values Indicate Statistical Significance at *p* < 0.05.

	ANCOVA		
Variables	F Value	df	*p* Value
Collection Timepoint	1.096	1.71	0.2986
Original Parasitoid Aphid host	0.6485	1.71	0.4223
Collection Timepoint × Original Parasitoid Aphid host	0.4947	2.104	0.6112
Choice Treatment	0.0729	1.71	0.7879
Honeydew Concentration (covariate)	4.9370	1.71	0.0295

**Table 4 insects-14-00010-t004:** Neither Timepoint nor Original Parasitoid Aphid Host Mediate Parasitoid Preference (Measured as the Relative Proportion of Time Spent on Each Treatment) for Grain Sorghum versus Johnson Grass Honeydew.

	ANCOVA		
Variables	F Value	df	*p* Value
Collection Timepoint	1.107	1.35	0.300
Original Parasitoid Aphid host	3.616	1.35	0.066
Collection Timepoint × Original Parasitoid Aphid host	1.331	1.35	0.256
Honeydew Concentration (covariate)	0.136	1.35	0.714

## Data Availability

Data regarding this study can be found at the Texas Data Repository (https://doi.org/10.18738/T8/DYI6PA, accessed on 20 December 2022).

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
