# Peer review of "Aphelinus nigritus* Howard (Hymenoptera: Aphelinidae) Preference for Sorghum Aphid, *Melanaphis sorghi* (Theobald, 1904) (Hemiptera: Aphididae), Honeydew Is Stronger in Johnson Grass, *Sorghum halepense*, Than in Grain Sorghum, *Sorghum bicolor"

_insects, 2022, doi:10.3390/insects14010010_

Round 1

Reviewer 1 Report

Review of Wright et al. 2022

This study provides some interesting findings regarding differences in honeydew composition between 2 host plants of the sorghum aphid. The chemical analysis appears reliable, although it is not very satisfactory in explaining the results of the behavioral assay, which produced somewhat ambiguous results.  The most important problem in the MS is the need to distinguish between 'attraction' (which would require the wasps to smell the differences at a distance) and 'retention' which is what the authors really measured in their preference test. They did not do any olfactometry, after all. The failure to make this important distinction colors the entire MS, especially the interpretation of the results. Just because wasps spent more time on leaves with one kind of honeydew vs another, does not mean they will be differentially attracted to it in a field situation, or even that they orient to the smell of honeydew at all (although they may, but it would be to volatile consituents).

The simple summary and the abstract are too similar. The authors have an opportunity to provide more information overall, especially in the Abstract, so no need to repeat things in both sections. The authors could go into more detail in the abstract about their findings. There 

There is a misconception, repeated several times in the MS, that honeydew is a resource necessary for egg maturation, when it is not. Host-feeding provides both the lipids and proteins required for egg maturation in Aphelinus spp. - honeydew is mostly just a source of energy for foraging, as explained in the Annu Rev reference that is cited. 

Finally, the inference in the final paragraph that honeydew "can recruit parasitoids" is not supported by any data presented here. This would imply attraction over some distance, which is more likely mediated by HIPVs emitted by infested plants.  There is also no mention of HIPVs anywhere in the Discussion, and this is the most likely reason for differential recruitment of parasitoids to sorghum vs JG - if in fact, this is the case (there is only 'unpublished data' cited in support of this contention). 

More detailed and specific comments are on the annotated pdf.

I also just realized an important potential confound in this experimental design. Because the honeydews was presented on its respective host plant leaves, responses to the leaves instead of (or in addition to) the honeydew cannot be ruled out! The honeydew should have been presented on a neutral substrate (e.g., filter paper). Most of the retention effect on parasitoids is likely due to the HD sugars, but still, any possible effect of the diff plants remains unknowable.

Author Response

Dear Reviewer 1:

Thank you very much for your review. In light of your suggestions, we have made the following changes:

The chemical analysis appears reliable, although it is not very satisfactory in explaining the results of the behavioral assay, which produced somewhat ambiguous results.

Our study ruled out the possibility of sugars or amino acids in influencing parasitoid honeydew preference. While the role (if any) of fumaric acid in mediating parasitoid preference still warrants attention, the chemical analysis has allowed us to explore other avenues affecting parasitoids (notably microbial metabolites or microbial volatile organic compounds).

The most important problem in the MS is the need to distinguish between 'attraction' (which would require the wasps to smell the differences at a distance) and 'retention' which is what the authors really measured in their preference test. They did not do any olfactometry, after all. The failure to make this important distinction colors the entire MS, especially the interpretation of the results. Just because wasps spent more time on leaves with one kind of honeydew vs another, does not mean they will be differentially attracted to it in a field situation, or even that they orient to the smell of honeydew at all (although they may, but it would be to volatile consituents).

We understand that retention more accurately describes our study and have replaced all references to attraction with parasitoid retention. This should clear any ambiguity with our results and discussion.

The simple summary and the abstract are too similar. The authors have an opportunity to provide more information overall, especially in the Abstract, so no need to repeat things in both sections. The authors could go into more detail in the abstract about their findings.

Due to the word limit, we were constrained in our ability to provide more detail. I have removed some sentences in the simple summary and reframed phrases in the abstract to better differentiate the two. In light of the first suggestion, I substituted all references to attraction with retention in both the simple summary and abstract.

There is a misconception, repeated several times in the MS, that honeydew is a resource necessary for egg maturation, when it is not. Host-feeding provides both the lipids and proteins required for egg maturation in Aphelinus spp. - honeydew is mostly just a source of energy for foraging, as explained in the Annu Rev reference that is cited.

We have removed all contextual linkages between honeydew consumption and egg maturation.

Finally, the inference in the final paragraph that honeydew "can recruit parasitoids" is not supported by any data presented here. This would imply attraction over some distance, which is more likely mediated by HIPVs emitted by infested plants. There is also no mention of HIPVs anywhere in the Discussion, and this is the most likely reason for differential recruitment of parasitoids to sorghum vs JG - if in fact, this is the case (there is only 'unpublished data' cited in support of this contention).

Instead of "recruitment", we have changed the phrasing to parasitoid "maintenance" i.e. our observations of parasitoid retention on Johnson grass SA honeydew suggest that we may maintain parasitoid populations on Johnson grass honeydew before the grain sorghum growing season.

I also just realized an important potential confound in this experimental design. Because the

honeydews was presented on its respective host plant leaves, responses to the leaves instead of

(or in addition to) the honeydew cannot be ruled out! The honeydew should have been presented

on a neutral substrate (e.g., filter paper). Most of the retention effect on parasitoids is likely due to

the HD sugars, but still, any possible effect of the diff plants remains unknowable.

More detailed and specific comments are on the annotated pdf.

In preliminary trials of these experiments, we tested parasitoid retention on Johnson grass and grain sorghum leaves containing a drop of water. This was specifically to rule out the effects of leaf texture/type on parasitoid retention. We observed no difference/preference for one leaf type over another and consequently proceeded with the experiments detailed in the manuscript. As the sorghum aphid is a cereal pest, we aimed to partially simulate what A. nigritus would encounter in the field. Therefore, we deliberately chose not to use filter paper. We have mentioned the lack of inherent parasitoid preference for grain sorghum or Johnson grass leaves in the discussion (lines 503-507).

We have received your annotated pdf and made the following adjustments:

Lines 13-14: “I don’t see how this follows, or is needed”

The goal of our experiments is to determine whether honeydew plays a role in retaining parasitoids. As the sorghum aphid is a cereal pest, this information is relevant to determine whether we could factor honeydew in a future biological control program. Hence, we included the “practicality of SA biological control” component.

Line 49-50: “You need to mention HIPVs here - a pretty important omission.”

We have updated the references to mention HIPVs

Line 61: “I don’t think there is any repellence to speak of. I think what you mean is that potentially unrecognized variation in honeydew quality/attractiveness as a function of host plant has not been explored.”

We removed the reference to repellence.

Lines 70-71: “So what? Don’t just list 4 references behind such a bland and pointless comment - tell us something interesting about what they showed!”

We removed this sentence and incorporated the references in the sentence before.

Line 139-140: “I would refer to ‘periods of accumulation’, rather than timepoints. Less ambiguous.”

We respectfully disagree. We explain how we define collection timepoints (for 24, 72, or 120 hours) immediately after introducing this term. We have added “after honeydew accumulated” before the “for 24, 72…” to further clarify the term. 

Line 208: “three accumulation periods”

We have kept the original term as per our response above.

Figure 1: “grass” – not capitalized

We have fixed this error.

Figure 2: “You need to provide a better description of Figure 2 as it is not a conventional representation. What is really being depicted here, and what do the elipses represent. The axes labels alone are ambiguous”

We have regenerated the figures and added the following to the caption: “Ellipses represent 95% confidence intervals. Triangle and circle points ordinated closer to one another are more similar.” In NMDS plots, the axes are not ordered and only exist to present the data in a way that highlights the most variability/dissimilarity among the variables. In other words, they are completely arbitrary.

Lines 432-433: “I don’t think you can talk about ‘recruitment’ because you haven’t measured attraction at any distance.”

We have changed “recruitment” to “foraging”

Line 445-449: “It also cannot be assumed that any such differences, if measured, would result in differences in honeydew sugars.”

We do not make this assumption. We only speculate that the two may be correlated. We were careful to use modal verbs in this section (“Higher honeydew sugar concentrations may result…comparisons between plant species might help explain our results”).

Line 453-455:

There seems to be a mistaken assumption here. The aphids are not feeding on the sugar - they are filtering out the nitrogen. Sugar is a waste disposal problem, so a higher sugar concentration is UNdesirable from the aphid perspective. That’s why they excrete it! You need to read chap. 2 of Aphid Ecology (Dixon, 2000).

Thank you for catching this error. We have now specifically mentioned nitrogen and suggested measuring amino acid content in phloem to help validate the compensatory feeding hypothesis.

Lines 462-463: You cannot say this. not attraction was measured - only retention on contaminated leaves.

We have modified the sentence to reflect retention, rather than attraction.

Lines 465 and 468-469: “OK, but only after contact with it….Again, the elicitation of search behavior after contact does NOT equate to its use as a cue for host location.”

We hypothesize that A. nigritus may use SA honeydew as a contact cue to forage for aphids. Contact/foraging cues have been reported in the literature (see Leroy et al. 2014 and Blande et al. 2008) without the measurement of volatiles emitted from honeydew. We modified the sentence to mention “foraging” as the type of cue.

Blande, James D., John A. Pickett, and Guy M. Poppy. "Host foraging for differentially adapted brassica-feeding aphids by the braconid parasitoid Diaeretiella rapae." Plant signaling & behavior 3.8 (2008): 580-582.

Leroy, PASCAL D., et al. "Aphid honeydew: An arrestant and a contact kairomone for Episyrphus balteatus (Diptera: Syrphidae) larvae and adults." European Journal of Entomology 111.2 (2014): 237.

Lines: 489-491: This is a bit missleading. Sugars are mostly used as a source of energy and can extend life. They can also *contribute* to ovigenesis in synovigenic parasitoids, but protein and lipids are far more limiting. That is why the adult females have to host-feed. Read your reference 66 more carefully. You can give them ad libitum sugar, but without host feeding, there will be no ovigenesis.

We have revised the statement to reflect sugar consumption as mainly an energy source.

Lines 498,517,549: “Such a myriad of things are going to affect parasitsm rates in the field, any connection to a ‘slight’ preference for one or other type of hoenydew is going to be very tenuous and speculative at best. I do not think such a connection can be justified.”

“I would not even speculate on factors influencing parasitism rates in the field. Too many factors, and your data are not sufficiently relevant.”

“Again, any direct inferences regarding pest management implications are, IMO, far to tenuous and speculative given the limiations of the data.”

We understand the complexity surrounding field parasitism and never assert that honeydew preference is the sole factor affecting parasitism rates. From our results, it is evident that preference does not contribute to observed field parasitism rates, at least not under the conditions of our experiments. To highlight the complexity of field parasitism, we acknowledge two other factors: leaf surface and parasitoid learning. We also raise the question of whether honeydew retention can lead to SA suppression, a topic for future studies in this space. We have added the following sentence to clearly acknowledge this myriad of factors: “Since the factors affecting field parasitism rates are complex in nature, differing A. nigritus parasitism rates may reflect physical instead of chemical plant traits.” In the last paragraph, we have emphasized “future honeydew-related pest management”, as our work aims to bring attention to one of the many factors affecting the possibility of SA biological control.

Lines 508 – 514: “But are sorghum leaves different in texture from JG leaves? And if the leaves themselves are affecting the preference, what does that do to your ability to make inferences about the relative preference for the honeydew?”

Mature sorghum and Johnson grass leaves differ in texture due to the presence of more bicellular trichomes on the latter species. Young leaves (like the three-week-old leaves used in our study) are relatively similar in texture. As mentioned above, we preliminarily tested A. nigritus preference for water deposited on grain sorghum and Johnson grass leaves but found no evidence of a leaf preference.

Once more, thank you for your constructive feedback, especially as it relates to the difference between attraction and retention.

Sincerely,

The Authors

Reviewer 2 Report

Dear all,

Your manuscript on parasitoid preference for aphid honeydew from different host plant sources was very well written and a pleasure to read. Overall, your results demonstrated that sorghum aphids feeding on Johnson grass had higher sugar concentration than those feeding on grain sorghum and that there was an increase in fumaric acid in the honeydew of the aphids feeding on Johnson grass. I found the fumaric acid a particularly interesting aspect of your results. Furthermore, the parasitoid attraction to Johnson grass honeydew, with the exception of the Johnson grass parasitoid at 24 hr, was very peculiar in light of the field observations, as you pointed out. I especially appreciated the discussion portion of this paper as it addressed all of my questions that arose while reading the results. 

Thank you very much for your work. The only suggestions I have are to add post-hoc information to the figure legends. Although, I don't think that is strictly necessary. Also, the suggested citation format on the Insects author guidelines recommends making the title of articles regular font and the journal name italicized. I found this to be the opposite in your article. This may help to be able to italicize the species names within the titles of articles. 

Author Response

Dear Reviewer 2: 

Thank you very much for your review. We are happy the manuscript made conceptual sense. The journal name should be italicized and the title in regular font to better showcase the species names. As the post-hoc figure suggestion was optional, we have left the figure labels as-is. However, we did regenerate the figures to improve the visual quality. 

Note: Due to a request from Reviewer 1, we have changed all references to parasitoid attraction. Instead we mention parasitoid retention on a honeydew surface to more accurately reflect our study.

Once more, thank you for the review.

Sincerely,

The Authors

Reviewer 3 Report

The manuscript is well written as evident in its readability and logical fluidity as one reads. The study was properly designed and executed. The manuscript is well referenced. Also, the results were well reported and explained in detail.

However, there are some minor revisions that I would recommend before publication.

Line 42: provide references

Line 64: provide references

Line 75: write the new scientific name of the aphid "Melanaphis sorghi"

Line 77: insert "Although some studies have documented a combination of management methods and production practices to manage this pest in grain sorghum (e.g., Seiter et al., 2019; Lahiri et al., 2021; Uyi et al., 2022), however, very little is known about the role of natural enemies of M. sorghi (Uyi et al., 2022; include other references if you will)”

Figures 2-4: Please improve the quality of the images

Line 397: delete " mature eggs" and replace with "assist with egg production and development"

Line 427:  insert "only" between "not" and "with" 

References:

Lahiri, S., Ni, X., Buntin, G.D., Punnuri, S., Jacobson, A., Reay-Jones, F.P.F., Toews, M.D, 2021. Combining host plant resistance and foliar insecticide application to manage Melanaphis sacchari (Hemiptera: Aphididae) in grain sorghum. International Journal of Pest Management 67, 10–19. 

Seiter, N.J., Miskelley, A.D, Lorenz, G.M.,  Joshi, N.K., Studebaker, G.E., Kelley, J.P.,  2019. Impact of planting date on Melanaphis sacchari (Hemiptera: Aphididae) population dynamics and grain sorghum yield. Journal of Economic Entomology, 112, 2731–2736. 

Uyi, O.Lahiri, S.Ni, X.Buntin, D.Jacobson, A.Reay-Jones, F.P.F.Punnuri, S.Huseth, A.S., Toews, M.D., 2022. Host plant resistance, foliar insecticide application and natural enemies play a role in the management of Melanaphis sorghi (Hemiptera: Aphididae) in grain sorghum. Frontiers in Plant Science 13,1006225.

Author Response

Dear Reviewer 3: 

Thank you very much for the review. In light of your suggestions, we have made the following modifications: 

Line 42: provide references

References have been provided.

Line 64: provide references

References have been provided.

Line 75: write the new scientific name of the aphid "Melanaphis sorghi"

The scientific name is now included.

Line 77: insert "Although some studies have documented a combination of management methods and production practices to manage this pest in grain sorghum (e.g., Seiter et al., 2019; Lahiri et al., 2021; Uyi et al., 2022), however, very little is known about the role of natural enemies of M. sorghi (Uyi et al., 2022; include other references if you will)”

This information has been included.

Figures 2-4: Please improve the quality of the images

We have regenerated and restructured the figures to improve the quality. We have also added color. 

Line 397: delete " mature eggs" and replace with "assist with egg production and development"

In light of Reviewer 1's comments, we have removed all references to egg maturation, including this line. 

Line 427:  insert "only" between "not" and "with" 

As we did not find a correlation between parasitoid honeydew preference and microbial composition, we have hypothesized that variations in parasitoid honeydew preference correlate with microbial metabolites or microbial volatile organic compounds, not with the microbial organisms themselves. To make this clearer, we have modified the sentence below:

"We hypothesize that the variation in parasitoid honeydew preference may not correlate with the presence of microbial organisms themselves, but rather with the differential presence of microbial metabolites or microbial volatile organic compounds (MVOCs)..."

Once more, thank you for the constructive feedback.

Sincerely, 

The Authors